# A Critical Review on Artificial Intelligence—Based Microplastics Imaging Technology: Recent Advances, Hot-Spots and Challenges

**DOI:** 10.3390/ijerph20021150

**Published:** 2023-01-09

**Authors:** Yan Zhang, Dan Zhang, Zhenchang Zhang

**Affiliations:** 1School of Materials and Environmental Engineering, Fujian Polytechnic Normal University, Fuzhou 350300, China; 2School of Big Data and Artificial Intelligence, Fujian Polytechnic Normal University, Fuzhou 350300, China; 3Fujian Provincial Key Laboratory of Coastal Basin Environment, Fujian Polytechnic Normal University, Fuzhou 350300, China; 4College of Computer and Information Sciences, Fujian Agriculture and Forestry University, Fuzhou 350002, China

**Keywords:** microplastics, imaging technology, artificial intelligence, science mapping, hot-spots analysis

## Abstract

Due to the rapid artificial intelligence technology progress and innovation in various fields, this research aims to use science mapping tools to comprehensively and objectively analyze recent advances, hot-spots, and challenges in artificial intelligence-based microplastic-imaging field from the Web of Science (2019–2022). By text mining and visualization in the scientific literature we emphasized some opportunities to bring forward further explication and analysis by (i) exploring efficient and low-cost automatic quantification methods in the appearance properties of microplastics, such as shape, size, volume, and topology, (ii) investigating microplastics water-soluble synthetic polymers and interaction with other soil and water ecology environments via artificial intelligence technologies, (iii) advancing efficient artificial intelligence algorithms and models, even including intelligent robot technology, (iv) seeking to create and share robust data sets, such as spectral libraries and toxicity database and co-operation mechanism, (v) optimizing the existing deep learning models based on the readily available data set to balance the related algorithm performance and interpretability, (vi) facilitating Unmanned Aerial Vehicle technology coupled with artificial intelligence technologies and data sets in the mass quantities of microplastics. Our major findings were that the research of artificial intelligence methods to revolutionize environmental science was progressing toward multiple cross-cutting areas, dramatically increasing aspects of the ecology of plastisphere, microplastics toxicity, rapid identification, and volume assessment of microplastics. The above findings can not only determine the characteristics and track of scientific development, but also help to find suitable research opportunities to carry out more in-depth research with many problems remaining.

## 1. Introduction

Plastic pollution is one of the most common problems which already threatens marine health, marine food, human health, marine tourism, and its associated climate change. There were 4.8 million to 12.7 million tons of plastic which entered the ocean in 2017 [1]. MPs (abbreviation for microplastics) is defined as “any synthetic or polymer solid particle with regular shape or irregular, size between 1 μm and 5 mm, and insoluble in water” [2] (p. 146). MPs are firstly and importantly derived from industrial production in specific application fields, such as cosmetics, pharmacology, textile industry, and medical diagnosis. A secondary source is the decomposition of larger plastic fragments into trillions of smaller fragments (MPs) [3,4]. The discovery of MPs in the marine food chain has raised more concerns about seafood consumption [5]. Such achievements have rapidly promoted the development of microplastic pollution research over the past few decades [6]. The different methods to quantify and identify microplastics were introduced in environmental matrices, including air [7], water [8], biota [9], and sediments [10].

AI-based (abbreviation for artificial intelligence-based) microplastic-imaging technologies are boosted by emerging cutting-edge technologies. They are series of new AI-based algorithms, new technologies, new applications that have a vital role in the related microplastics fields. They have grabbed very wide attention due to the following benefits: improving operational efficiency, reducing time-consumption effectively, subverting the existing imaging technologies, facilitating quantification methods, and providing new cognitive thinking. The key technologies of AI-based microplastic-imaging technologies include smart Unmanned Aerial Vehicle (UAV) [11,12,13], artificial intelligence technology, and intelligent robot technology [14]. Driven by these new AI technologies, the traditional way of the microplastics debris imaging technology research has been completely transformed.

Furthermore, compared with large-size plastics, the image classification, identification, and quantitative research of microplastics is a complicated scientific field due to their small size and huge volume, so scholars have performed many studies involving theories and applications on AI-based microplastic-imaging technologies. Accordingly, the application of AI-based microplastic-imaging technologies has been involved in many areas, such as deep learning model [15,16], environment monitoring [17,18], microbial communities [19,20], microplastics quantification—identification technology, classification technology, and automatic quantification [21,22,23,24,25,26].

In order to explore the disciplinary boundaries and research paradigms of microplastic imaging technologies, the current research circumstances and development hot-spots were investigated based on the current published literature in the field of artificial intelligence-based microplastic-imaging technologies. A total of 97.07% of all the published literature was published after 2019 (based on Web of Science (WoS)). This explosion of growth and benefits on related microplastic-imaging research activities have driven us to perform a thorough and objective analysis, including the current challenges and opportunities which the field of AI-based microplastic-imaging technologies obtained. It is very crucial for scholars interested in AI-based imaging field to carry out better and more in-depth research. Moreover, the characteristics and track path of technology development can be revealed from the perspective of visualization through science mapping analysis [27,28]. Additionally, this paper examined hot research themes and recent advances, discussed the challenges, and predicted the future trends with many problems remaining.

Science mapping analysis, from a statistical and quantitative analysis perspective, conducts visualization analysis via science mapping visualization tools, for instance CiteSpace and VOSviewer and Scimago Graphica [29,30,31], therefore facilitating the understandability of analysis results. Text mining and visualization in the scientific literature has been deeply engaged in different research fields, such as Blockchain Research [32], Decision Ambiguity [33], Advanced Deep Learning Research [34], Humanities and Social Sciences [35], Business and Economics Research [36], journals of Operational Research [37], Information Technology and Sciences [38], and the Environment field [39]. The three science mapping tools, which are CiteSpace and VOSviewer and Scimago Graphica, can facilitate the science mapping analysis in discovering intellective bases and research frontiers of AI-based microplastic-imaging technology from a different point of view. For instance, the co-occurrence network analysis of keywords determines the main research topics; the co-operation network, such as co-countries, co-authors, and co-institutes, can reveal the top influencing objects; the timeline view and overlay journals analysis can show some variation over a certain period of time based on research keywords or themes. Such a methodology is named science mapping [31,40].

In this research, according to core articles and proceeding papers from 2018 to 2022 in the past five years, science mapping was introduced to assess the current state of research on AI-based microplastics imaging. As of now, there is no comprehensive summary on this research in the recent published literature. This review will help to fill in this gap and cover the related published key research results and highlight related technical issues and future research prospects. The contributions of this research can be summarized in the following points: (1) Illustrate the fundamental characteristics of AI-based microplastic-imaging documents, including the types of publications, temporal distribution of publications, popular research areas, top occurrences keyword and keyword density mapping; (2) Explore productive or active countries/regions, institutions, and authors from a macro- to micro perspective, and plot the cooperation relationship networks to point out the relatively active authors, institutions; (3) Additionally, determine the citation and co-citation structure networks on authors and references and also clearly show the research hot-spots and knowledge flow of all AI-based microplastic-imaging publications based on overlay analysis, temporal evolution, and cluster view; (4) With the current research context, the challenges and problems should be taken into consideration which AI-based microplastic imaging faced; and (5) Suggesting the future prospects in this field. Hopefully, this research result will provide a useful reference for related scholars and further applications of AI-based microplastic imaging or the other fields.

## 2. Methods

### 2.1. Data Acquisition and Whole Process

The research data were acquired from the Web of Science (WoS) Core Collection of American Institute of Scientific Information (ISI). The WoS is regarded as the most comprehensive, containing the most authoritative literature in its dataset. On 31 May 2022, all the literature related to AI-based microplastic-imaging technology was searched and further investigated in this study.

As mentioned in the introduction, science mapping is used to reveal and visualize the development trends and movements in a particular research field [41]. This paper introduced related text mining and visualization methods to evaluate the development problems and research hot-spots of AI-based microplastic-imaging documents. CiteSpace and VOSviewer and Scimago Graphica, as three popular science mapping tools, effectively expose the interrelationship of the literature and visualize interaction from different views, such as clustering and dynamic timeline [29,30,42]. Through several science mapping tools and visual mining, containing co-occurrence network analysis, timeline view analysis, overlay journals analysis, and citation literature analysis, this research reviewed the co-occurrence author keywords, citations analysis, co-operation network structure analysis, hot-spots, and emerging problems on AI-based microplastic-imaging research.

The whole framework of science mapping in the AI-based microplastic-imaging field can be illustrated in Figure 1. The following knowledge mapping analysis of AI-based microplastic-imaging technology consists of five steps as shown in Figure 1: the first is to obtain the data from the WoS dataset with the assumptions and conditions; the second step is data collection to acquire and save the key literature into MySQL and txt format file; the third step is that three science mapping tools are engaged to assist; the fourth step is data visualization with indicators and methods, and the last step is mining and analysis to obtain recent advances, hot-spots and problems of AI-based microplastic-imaging technology.

### 2.2. Assumptions and Conditions

Three parameters, search terms, country names, and database time-range, were pre-set for further searching, given as follows:

#### 2.2.1. Search Terms

The below search terms in Table 1 were combined then inputted into the search engine in the Web of Science website, which found out the publications involving these searched items to appear in the related publications.

#### 2.2.2. Country Names

The geographic location information of the article is determined by the address of the author’s research institution. Among these publications, the literature marked from England, Scotland, Northern Ireland, and Wales was defined as the United Kingdom (UK), at the same, literature from Mainland China, Hong Kong, Macao. and Taiwan was regarded as China, individually in Table 2.

#### 2.2.3. Database Time-Range

The review data were acquired from the Web of Science (WoS) Core Collection of American Institute of Scientific Information. In addition, there are 5 categories, Science Citation Index Expanded (SCI-EXPANDED), Social Sciences Citation Index (SSCI), Conference Proceedings Citation Index—Science (CPCI-S), Conference Proceedings Citation Index—Social Science and Humanities (CPCI-SSH), and Emerging Sources Citation Index (ESCI) as our research data. The related database time span is as below in Table 3.

### 2.3. Science Mapping Analysis Tools

The map of scientific knowledge is a graph which can visualize the relationship and interaction between the movements and interaction of scientific knowledge. The research of scientific knowledge map is based on scientific knowledge, which belongs to the category of scientometrics and involves the cross fields of science, such as applied mathematics, information science, graph theory, and information metrology. A multiple-perspective visualization analysis method facilitates analytic and sense making tasks to analyze recent advances and hot-spots.

VOSviewer is a science mapping tool to help construct and visualize bibliometric network relationship. These network relationships can be visualized to reveal co-occurrence networks, citation network relationships, and density mapping relationships of important terms retrieved from the searched literature.

Scimago Graphica is a good visualization tool about the geographic location information of the literature marked by the author’s address to identify the cooperation among countries/regions distribution.

CiteSpace software used in this paper is proposed to explore the key path of the evolution of the field of knowledge by integrating network visualization, spectral clustering, automatic cluster labeling, timeline view, and dual over-lay journeys based on citation analysis.

## 3. Fundamental Characteristics of AI-Based Microplastics Imaging Publications

This section presents the fundamental characteristics of the AI-based microplastic-imaging literature, containing the types of publications, temporal distribution of publications, popular research areas, and keyword density mapping, which will help to know more about the latest situation in the area.

### 3.1. Analysis of Literature Types

For the 69 searched publications in AI-based microplastic-imaging field, the first document was written by Lorenzo-Navarro, J and Castrillon-Santana, Modesto (2018), that is, Automatic Counting and Classification of microplastics Particles, a proceedings paper. The types of the 69 documents about AI-based microplastic-imaging technologies are presented in Figure 2.

From Figure 2, the main type of literature is articles, with 54, which account for about 78%. The second is the type of proceedings papers, accounting for about 16%; three review articles, accounting for 4.35%; one (1.45%) is early access.

### 3.2. Number and Proportion of Published Papers per Year

The number and proportion of published papers per year since 2012 are depicted in Figure 3. From 2012 to 2017, the annual growth records remained zero. From the graph, the number of publications grew gradually since 2018. Especially for 2021, the publications increased dramatically, and there are 31 publications, accounting for about 45%, which are more than the total of the past three years from 2018 to 2020. This fully explained why the research on AI-based microplastic imaging is becoming an increasing hot-spot recently and the crossing fields may obtain more findings and achievements.

### 3.3. Top 10 Research Areas

The top ten research themes of the documents are shown in Tree map Figure 4. Environmental Sciences Ecology are the most active research area. The total number of documents for Environmental Sciences Ecology is 37, accounting for 53.62%. The second is Engineering area with 14, which is about 20.29%. The third is Computer Science with 10, accounting for 14.49%. Furthermore, research is widespread in Optics (10), Chemistry (8), and Water resources (4). AI-based microplastic imaging involves various areas and drives tremendous development in many research fields.

### 3.4. High Frequency Keywords and Density Mapping

Based on the statistics about the keywords from 2018 to 2022 among 69 publications, there were 357 keywords, where 6 keywords with more than 10 times in Figure 5. Microplastics (42), machine learning (29), deep learning (17), identification (13), pollution (13), and classification (12) are the top sic keywords in the research high frequency themes in Figure 5, which means artificial intelligence technologies have already been engaged in the research direction of microplastic-imaging technologies to a very deep level.

Keyword heat density map helps to visually infer the concentration of keyword research by density, overlap, etc. In order to more easily identify the current spatial distribution of research areas, it can visually gain insight into density data (i.e., color depth) to detect topic heat, distribution status, and help turn curiosity about this area into insight. Visualized from VOSviewer, the themes of the 69 documents are revealed in Figure 6. In addition to microplastics, there are some keywords, such as machine learning, deep learning, classification, abundance, identification, and artificial intelligence, with higher frequency, which are all colored in red. The map of Figure 6 provides another evidence about the research directions from density mapping perspective and also helps to understand more where the focus and research directions are in the AI-based microplastic-imaging technologies.

## 4. State of AI Applications in Microplastics Imaging Research

### 4.1. Most Active Object Analysis and Collaborative Network Analysis

The sub-section shows the most active and contributing countries, research institutes, and authors from macro to micro perspective. Mastering the partnership among them helps pinpoint the most active or contributing or effort institutions and authors and also will help to know more about who or which scientific research institution is relatively active in the research hot-spots.

#### 4.1.1. The Collaboration Network of Countries/Regions

As mentioned in Figure 7, China is the most active country, accounting for 20 from 2017, followed by Italy with 9. Then, the third to USA (six), then Australia (five), Spain (five), Canada (four), Germany (four), Brazil (three), and UK (three). China and USA have the same total link strength with other countries (eight). Followed by Germany, Czech and Malaysia with five. Then, the third to Switzerland and Philippines (four), then Spain and Canada (three). There are a total of 31 countries/regions, which are presented via the closest cooperation network in Figure 8. To quickly identify the most cooperation distribution of different countries, density map is used to visualize national cooperation distribution density. The greater the node weight of national cooperation, the greater the surrounding density.

The countries/regions mentioned in Figure 8 are separated into 16 groups, and various colors stand for specified countries/regions. The circle-nodes represent different countries/regions, and the sizes of the circle-nodes reflect the total number of publications individually. The connections between two circle-nodes show that there are collaborative relationships among them. The thicker the connection line, the closer they work together. In each group the most collaborative countries/regions are Malaysia, USA, Germany, China and Czech and their total link strength is over 5. However, another 10 countries (such as Japan, India, and South Korea) did not have the collaboration between any 2 countries and their total link strength is zero. At the same time, the United States and Germany have a thicker line connection, marking a closer cooperation between them; similarly, China and Australia have a thicker line connection, representing a close cooperation between the two of them in research. Other than that, the nodes between each country are slim, which means that cooperation between them is sparse. Further, the geographical distribution shows that the regions where more research is conducted are concentrated in developed countries and regions or larger developing countries, and relatively less in South America and Africa. In addition, China and USA and European countries, three of these countries/areas should promote cooperation urgently to strive for more scientific breakthroughs. In addition, in Africa and south of America a few countries carry out related AI-based microplastic-imaging research and better collaborate with other developed countries or areas.

#### 4.1.2. The Most Contributing Affiliations

Figure 9 presents the most contributing research institutes. According to Figure 9, the consiglio nazionale delle ricerche cnr has the largest total number of publications with seven publications. The istituto di scienze applicate e sistemi intelligenti (six), university of hong kong (five), universidad de las palmas de gran canaria (three), and university of naples federico II (three) follow.

From the perspective of total citations, the university of toronto occupies the most contributing institution with 47, and the second through fifth highest contributing research institutions are the icar central inland fisheries research institute (46), indian council of agricultural research icar (46), consiglio nazionale delle ricerche cnr (41), and istituto di scienze applicate e sistemi intelligenti eduardo caianiello isasi cnr (41). The situation motivates us to further examine the collaborative relationship among these more contributing research institutes and help understand more which institute is popular in the related AI-based microplastic-imaging research area.

#### 4.1.3. The Cooperation Networks of All Institutions

There are 133 institutions in the field of AI-based microplastics imaging. Figure 10 denotes the collaborative relationship network structure for 133 institutes, among of which there are the tightest collaborative network with 13 institutes. From Figure 10 the node size represents the total number of published documents by the related co-operation institutions. The gray nodes present no partnership between the institutions that published any document. There is no single organization or individual can stand alone in the face of new microplastic pollutants and must seek cooperation to face the environmental crisis. In order to improve the poor relationship situation, collaboration dramatically has to be strengthened for these isolated institutions so as to contribute more and remove “data island” in further research area.

#### 4.1.4. The Most Active Authors

Using Sankey diagrams, it is very helpful to show the correlation between active authors (number of pieces of literature), authors’ countries, and authors’ scientific institutions. To a certain extent, it can reflect the level of author activity. Based on the results of this statistics from Figure 11 between 2018 and 2022, 356 authors were involved in research area over the past five years. The most active authors were involved in the research area from Figure 11. According to the statistics, Bianco V is the most active author from Consiglio Nazionale delle Ricerche (CNR) in Italy who published six articles with H-index (3). In addition, from Figure 11, we can see further that Ferraro P., Memmolo P., Carcagni P. and Distante C. from the same institute called Consiglio Nazionale delle Ricerche (CNR) which is the same one with Bianco V’s and also there are still three authors, Lam E.Y., Yeung C.H., and Zhu Y.M., from the same institute called University of Hong Kong. This phenomenon revealed that the researchers from the same institute are easier to carry out research and cooperation internally in the relevant fields.

#### 4.1.5. The Closest Cooperation Network of Authors

As shown from Figure 12, we can see that there are some hot-spot areas which there are several authors with a close working relationship. In addition, there are too many gray nodes which indicate that the authors did not have any cooperation or link with other institutions. According to the above visualization analysis, different colored clusters almost have their own close cooperation and the related authors from the same institution just only worked together and performed the research internally, but for different institutions, the authors missed the cooperation in the network where there is no link or connection between different colored groups which called “Data Island”. It can shed some lights from this phenomenon that the idea of collaboration deserves to be promoted and strengthened in different institutions among different countries attributed to facing new environmental problems.

#### 4.1.6. Co-Occurrence Author Keywords

Keyword analysis is very helpful in identifying trends and trajectories in any field [43]. Keywords are very convenient to show on a topic of a specific field of research and help highlight the active themes in the document [44].

Figure 13 reflects the co-occurrence network from author keywords’ perspective in the research area. The keyword co-occurrence network diagram can give a correlational association that exists between several keywords, and this association can be expressed in terms of the frequency of co-occurrence. It is generally believed that the more the number of occurrences of a word pair in the same document, the stronger the relationship between the two topics. A total of 357 author keywords were retrieved by setting the least parameter to 2 and joining deep learning, machine learning, classification, and artificial intelligence into microplastic-imaging field.

From Figure 13 the node signifies the keywords of authors, and the node-size represents the occurrences. Each connection has a strength, which is the Links attribute and the Total link strength attribute. The larger the value, the stronger the connection strength. In the visualization of a network map, nodes with higher weights will be displayed larger than nodes with lower weights in a cluster. Nodes, which stand for different author keywords, may be put into various groups. A cluster consists of related nodes around a theme in a map. Specifically, the link of “microplastics” is 73 and the second is machine learning with 62, queued by deep learning with 49, classification with 38, artificial intelligence with 22. Therefore, it roughly and vividly characterizes the topic keywords mapping network around the AI-based microplastic-imaging technology from the perspective of keyword co-occurrence network.

### 4.2. Citation and Co-Citation Structure Analysis

In order to analyze the activity of the citation authors and references deeply, this sub-chapter carries out cited analysis and co-cited analysis from the perspective of authors and references. Thus, scholars interested in AI-based microplastic-imaging technology field can pinpoint the relatively well-referenced literature and academics exactly. Citation analysis uses various statistical methods, mathematics and other tools to reveal the status quo, internal correlation of the existing knowledge base in the AI-based microplastic-imaging field.

#### 4.2.1. Citation Network Structure Analysis of Authors

According to VOSviewer, the citation network is presented in Figure 14. From the citation network structure, 43 of 365 citation authors construct the closest relationship network. In addition, these 43 authors are grouped into five clusters. Nodes and their sizes represent authors and their citation level, individually. The larger the node, the more the authors are cited. Lam Edmund y, Zhu Yanmin, Bianco Vittorio, and Ferraro Pietro in the citation network have the larger sizes of nodes.

For detailed information, there are the most cited authors based on some important indicators, containing Total Publications, Total Cited, Links, and Total Link Strength by setting the threshold of 2 in Figure 15. Based on the results of Figure 15, in these involved authors Ferraro, Pietro, Bianco, and Vittorio are the most active through 6 publications and 41 citations from the same institute called Consiglio Nazionale delle Ricerche (CNR). Ferraro, Pietro and Bianco, Vittorio focused on researching the microplastics including microplastics identification, classifying and automatic detection via holographic imaging and machine Learning [45,46,47].

#### 4.2.2. Co-Citation Network Structure Analysis of Authors

In Figure 16, 162 of the 2158 co-cited authors construct the tightest co-citation relationship which are separated into five clusters with different colors (red, blue, purple, amber, and green). The link between two co-cited authors denotes both appeared in the same document at the same time. The stronger the line, the more often two authors are co-cited.

Figure 17 details the most co-citation authors according to related indicators, such as number of citations, links, and the total link strength.

Additionally, Primpke S. is the most co-citation author from the co-citation network with citation of 40, which 278 authors cited. Primpke S. specializes microplastics classification and imaging, critical assessment, and identification [48,49].

Furthermore, a citation burst can help point out a burst object within a time period which is associated with a surge of citations. In Figure 18, from 2018 to 2022, Simonyan K is the best citation burst author between 2020 and 2022 from bibliometrics perspective.

#### 4.2.3. Citation Network Structure Analysis of References

From Figure 19 combined with Figure 20, it clearly shows that 34 of 69 documents constitute the cited network of references. The most active cited references are Martin (2018) with 69, which is marked the most size of node with different colors (blue, red, amber, purple and green) for different cited reference networks in Figure 19. The second is Fallati (2019) with 51, followed by Cowger (2020) with 47, Ng (2020) with 34, and Guo (2020) with 34.

#### 4.2.4. Co-Citation Network Structure Analysis of References

The co-cited references around AI-based microplastic-imaging technologies were studied in a mapping. Based on VOSviewer by setting threshold to five, 61 of 2655 cited references network with three main groups (red, blue, and green) is visualized in Figure 21. Coupled with Figure 22, one reference has the strongest citation with 20, i.e., plastic waste inputs from land into the ocean. It provided an estimation about the mass of land-based plastic waste entering the ocean [50], and was published on Science in 2015, followed by Hidalgo-ruz et al. [49] and Primpke et al. [48] with citation 18 and 14. As shown in Figure 22, the most active co-cited references can be reflected that all of them constitute the intellective base to understand and investigate further in the microplastics field.

## 5. Research Hot-Spots and Development Trends

The use of AI methods to revolutionize environmental science was progressing toward multiple cross-cutting areas, dramatically increasing aspects of the ecology of plastisphere, microplastics toxicity, rapid identification, and volume assessment of microplastics, which help to master the research hot-spots, knowledge flow and development temporal evolution on AI-based microplastic-imaging technologies in this section.

### 5.1. Knowledge Flow Analysis Based on Overlay Journals

The overlay journal view is engaged in our knowledge flow analysis, which helps reveal which knowledge domains currently construct the knowledge base for the current research or research front. From the perspective of the areas in which the existing literature is located from Web of Science database, which existing knowledge base has a large impact on the current research frontier, and which has less impact, or even no impact. Thus, research areas for further collaboration or strengthening can be derived from the journal overlay view.

In order to describe the knowledge flow of the publications more visually, the overlay journal analysis was discussed [39]. The mapping reveals quite a few things clearly, for example, which journals are most popular in terms of how frequently they publish papers on the topic. The overlay journal graph consists of two parts, such as citing side in the left side, and cited side in the left side. The curved lines signify the relationship between the citing side and cited side.

From the perspective of citation, journals published on the left side and journals cited on the right side are relatively concentrated. To make it easier to understand, the red arrow with knowledge workflow direction in Figure 23 indicates that the current literature knowledge comes from the knowledge domain of the cited journal on the right side. On the left about citing journal map, the research themes mainly fall into group 1 with mathematics, systems, mathematical, group 3 with ecology, earth, marine, group 5 with physics, materials, chemistry, and group 7 with veterinary, animal, science. Accordingly, for these research themes the total number of authors is enormous. On the right side about the cited journal map, the research literature covers a lot of areas, such as system, computing, computer in group 1, environment, toxicology, nutrition in group 2, chemistry, materials, and physics in group 4.

From the perspective of knowledge exchange, the knowledge flow mainly flows from journals such as chemistry, materials, physics, environment, toxicology, nutrition, system, computing, and computers to the citing journals such as mathematics, systems, mathematical, ecology, earth, marine, physics, materials, chemistry, veterinary, animals, and science.

By contrast, the research themes in AI-based microplastic-imaging technology fields have played an important role in various areas, and continue to expand its influence in future days. However, for now, it can also be seen that the current AI-based microplastic imaging technology is relatively concentrated, and most areas are not yet covered in depth, indicating that the current research is not extensive enough.

### 5.2. Keywords Network and Their Temporal Evolution

A color temporal bar is located at the bottom of the graph in the visualization in Figure 24. The color temporal bar expresses how occurrences are mapped to colors based on related keywords from May 2019 to May 2021. In this temporal evolution graph, keywords dyed in cool tones signify a relatively early year of research activities, conversely, keywords dyed in warm tones indicate a relatively late year of research activities. The whole figure reflects the temporal evolution of keywords over time in the AI-based microplastic image technology field over the past five years.

Based on the text box color, the overall time period is divided into two time periods, roughly. The former period is the microparticles and classification from May 2019 to May 2020, and the main focus is on microplastics particles, microplastics pollution and image classification using some optical instruments and devices, UAV, etc. Based on artificial intelligence technologies, the second half was from May 2020 to as of now, scholars devote themselves to the quantitative effects of microplastics, automation, identification in different environment, such as marine, surface waters and climate change via machine learning, deep learning, and neutral network algorithm. To a certain extent, artificial intelligence technology promotes the related microplastic-imaging technologies rapidly and in future it should still play an important role in microplastic-imaging direction.

### 5.3. Research Hot-Spots Analysis of Cluster View

The cluster view in the CiteSpace system can reflect the distribution of research fields from different perspectives. Under the cluster visualization graph of AI-based microplastic-imaging technologies in Figure 25, 7 clusters are separated and marked with #0–#6 with different colors (red, green, amber, dark blue, cyan blue, orange, and purple), which mainly occurs from 2018 to 2022.

By reviewing the development history of artificial intelligence in microplastic imaging all over the world, it can be found that with the rapid development of artificial intelligence technology, machine learning, especially deep learning, is widely introduced into various research areas, such as imaging, recognition, classification, and quantification of microplastics imaging. Due to its powerful ability to generalize features from data rather than manually identify them based on domain-specific knowledge, deep learning has rapidly become a mainstream artificial intelligence technique in the past few years and has significantly improved the performance of microplastic-imaging applications. From the cluster view, there are seven clusters (#0 deep learning, #1 environmental monitoring, #2 microbial communities, #3 classification technique, #4 automatic quantification, #5 identification technologies and #6 other research trends) which would be analyzed further.

The #0 cluster deals with deep learning research in anthropology marine debris and microplastics, with research subjects mainly about deep learning algorithm, artificial neutral network, transfer learning, microparticles, and k-nearest neighbor (KNN) mode. From 2018, scholars paid attention to the critical keyword of deep learning in environmental microplastics fields and up to now too many scholars treat deep learning model and algorithm combined with microplastics fields as a hot-spot research direction and also in this research direction there are too many scholars and fruitful achievements. With development of deep learning algorithm and computer vision and artificial intelligence, scholars introduced deep learning algorithms to improve to monitor or detect or classify micro-plastics. Manifold Embedded Distribution Alignment (MEDA) transfer learning algorithm as modelling method in combination with the ultra-portable Near-infrared (NIR) sensor was a promising solution for low-cost and efficient field detection of plastic contaminated level in soil [51]. A deep-learning method was demonstrated for the removal of instrumental noise and unwanted spectral artifacts in Fourier transform infrared (FTIR) or Raman spectra, especially in automated applications in which a large number of spectra have to be acquired within limited time [52]. A robust classifier based on k-nearest neighbor (KNN) model was innovatively proposed to differentiate the chemical types of environmental MPs samples to classify the environmental MPs and effectively eliminate the interference of spectral distortions and diversity [53]. A machine learning model combined image analysis of fluorescent particles with classification models was proposed to detect and identify particles spiked in marine environmental matrices in a straightforward, cost- and time-effective yet reliable way [54]. A machine learning algorithm, based on k-nearest neighbors (KNN) classification was used to efficiently identify FTIR spectra of classical polymers such as poly(ethylene) in a fast and reliable automated way [46].

The #1 cluster concentrates on the environmental monitoring on marine debris pollution. From 2018 marine microplastics pollution has already attracted more and more attention so that scholars have increased the monitoring of marine microplastics particles, which is attributed to the large scale of microplastics. In the past 5 years, microplastics environmental monitoring is always another hot-spot research direction. Garbage classification by aerial images manual and automatic processing through machine learning are reliable and results justify the implementation [11,55]. The state-of-the-art deep-learning-based autonomic supervisory control system containing optimally smart robots works well for monitoring underwater ecosystems and marine debris to acquire underwater sea life and debris floating on the ocean surface [56]. With an unmanned aerial vehicle (UAV) and deep learning computational methods, monitoring a wide area at a low cost in a standardized was introduced to estimate the abundance and area of marine debris coverage and also related hotspots where marine debris accumulates [57].

The cluster #2 reveals microbial communities to explore and study the environmental impact of a range of microplastics and microplastics water-soluble polymers. From cluster #2 timeline view, especially in the development of artificial intelligence technology, the impact of microplastics and polymers on the environment has increasingly become a research hotspot. It includes research subjects of recognition, bioaccumulation, base line correction, plastisphere community, algae, aggregation, and food quality. Biomarkers of the plastisphere were studied using random-forest machine learning about the impact for the microbial ecology of the new anthropogenic ecosystem—plastisphere and explored environmental drivers of the plastisphere community variation in the freshwater and seawater ecosystems [58]. To better understand the hazardous effect of micro-plastics, in vivo and in vitro toxicity database and deep learning artificial neural network models combined approach is appropriate to provide insight into the toxicity mechanism of the broad range of environmental chemicals, such as plastic additives [59]. Machine learning algorithm revealed a close association between microplastics content in fishes and surface water, indicating risk associated with floating microplastics to the aquatic biota for occurrence, fate and removal of microplastics as heavy metal vector in natural wastewater treatment wetland system [60]. A multi-feature superposition analysis boosting (MFAB) machine learning (ML) approach identified and predicted the importance, interaction networks and superposition effects of multiple features about microplastics pollutants on realistic environments in complicated climatic and geographic scenarios, overcoming the bias from general studies [61]. Based on machine-learning prediction MP size is the most critical factor that should be considered in future laboratory tests and eco-toxicological risk assessments for microalgae [55].

The #3 cluster offers a series of classification technique and exploratory research. It includes research subjects of transfer learning, image processing, classification tree, analytical model, machine learning, and automatic identification. Machine learning models have always been treated as a research hotspot especially for microplastics classification technique recently. In the past five years, scholars have already explored and studied different microplastics classifier methods and models in combination with specific scenarios to detect and classify microplastics. The holographic coherent imaging approach based on machine learning (ML) is able to identify microplastics independently from their morphology, size, and different types of plastic materials, thus boosting the classification performance and reaching accuracy higher than 99% in classifying thousands of items [45]. A new approach multiple fluorescence signals from the sensor via supervised machine learning, which specifically or nonspecifically interacted with the polymers was applied for polymer classification for next-generation sensing systems in wastewater or natural environments [62]. A polarization-resolved holographic flow cytometer in a Lab-on-Chip (LoC) platform was engaged to add material specificity while operating in a microfluidic stream modality in classifying natural and microplastics fibers through a machine learning numerical pipeline [63,64]. A new approach for the classification of microbeads (MBs) based on microscopic images via a Convolutional Neural Network (CNN) was introduced to classify, and characterizing microplastics, which achieved a classification performance of 89% for microbeads (MBs) in wastewater [64].

The #4 cluster reveals automatic quantification technologies of microplastic-imaging segmentation. Due to the large-capacity characteristics of microplastics particles, artificial intelligence technologies such as convolutional neutral network, image segmentation, are used to automatically identify and count microplastics quickly and accurately, which is still a hot-spot research direction for scholars from all over the world. An ad-hoc methodology for monitoring and automatically quantifying Anthropogenic Marine Debris (AMD), based on the combined use of a commercial Unmanned Aerial Vehicle (UAV) and a deep-learning-based software (such as PlasticFinder) was demonstrated—for the first time—the potential of deep learning for the automatic detection and quantification of Anthropogenic Marine Debris (AMD) [65]. High-performance segmentation and shape classification based on deep learning (U-Net and ResUNet) were engaged in scanning electron micrographs of microplastics particles (fragments or beads) in the range of 50 μm–1 mm and fibers with diameters around 10 μm with high accuracy, which is remarkably cheaper and faster than manual labor [66]. A high throughput screening method based on near-infrared hyperspectral imaging (HSI-NIR) was proposed to identify microplastics in beach sand automatically with minimum sample preparation using multivariate supervised soft independent modelling of class analogy (SIMCA) classification models [67]. A U-Net neural network was trained to segment microplastics and image post-processing techniques were then applied to count the number of microplastics as well as highlight their position in an image for the automated counting of microplastics [68]. The use of Kernel ridge regression-based machine learning to estimate the number of microplastics particles on the basis of aggregate particle weight measurements is better at predicting the counts of larger and more homogeneous samples [69].

The #5 cluster explores several new methods about microplastics identification technologies combined spectroscopic techniques with machine learning. The research subjects are mainly about infrared spectroscopy, deep learning, microscopy, water monitoring and FTIR. ATR–FTIR, NIR reflectance spectroscopy, and LIBS coupled with machine learning classifiers can be used to identify both consumer and environmental plastic samples of plastic-type identification and characterization rapidly [70]. Artificial intelligence-enabled coherent imaging holographic technologies was engaged in identifying and mapping the microparticles content of marine waters so as to unlock new possibilities in the fields of diagnostics and environmental monitoring [71]

The #6 cluster presents other research trends around related microplastics pollution. The research subjects are mainly about prediction, deep learning, image dataset, and surgical waste. Particle and salinity sensing for the marine environment via deep learning using a Raspberry Pi was proposed to identify mixtures of particles in a solution via analysis of scattered light to demonstrate a portable and low-cost environmental marine sensor technology [72]. An improved random forest machine learning regression model to the observed litter concentrations was investigated in which environmental variables play an important role in the beaching process and exploring the variability of beach litter concentrations and the related further finding is that tides play an especially important role, where an increasing tidal variability and tidal height leads to less litter found on beaches [73].The t-distributed stochastic neighbor embedding machine learning algorithm revealed a strong association between microplastics abundance with turbidity, phosphate, and nitrate, which was having comparable microplastics removal efficiency with previously reported advanced way [60]. Since these types of surgical masks and gloves waste are scattered around us and turned into microplastics during the pandemic of COVID-19, different versions of the You Only Look Once (YOLO) are applied as the architecture of a computer vision-based system for surgical waste detection [74].

In Figure 25, all of author keywords are separated into 7 clusters, such as environmental monitoring, microplastics, deep learning, machine learning, unmanned aerial vehicles, shape classification, automation, and chemometrics. Cluster #0 is the category with the longest time. There are four hot-spot research fields which are machine learning, microplastics, convolutional neural network and deep learning as shown via tree ring history style in Figure 25. The vast majority of keywords broke out with the development of artificial intelligence, such as image classification, size, quantification, counting, identification, recognition, and image processing. With the development of IT, special for advanced artificial intelligence technologies, it rapidly promoted artificial intelligence-based microplastic imaging in different research directions. Especially, deep learning is another emerging hot-spot in the last few years again after machine learning, and the phenomena obtained more attention from scholars and scientific research institutions. Therefore, it is particularly important to examine the current research achievements in a timely manner, which emphasizes the significance of this review once again.

## 6. Discussion

The research hot-spots in the field of AI-based microplastic imaging are mainly concentrated in specific practical technologies and the role of the microplastic-imaging fields is emphasized. AI-based microplastic imaging is the integration of microplastic imaging and artificial intelligence technology and environment science, the latter promotes the development and progress of the former dramatically. Consequently, in the next period, due to AI methodology development with many problems remaining, how to jointly achieve deep cross-domain innovation between MPs imaging and AI will be a very significant challenge. Several issues should be seriously considered, such as depth and interpretability of algorithmic models, cooperation features, data open access, the and data complexity from huge-volume imagery.

### 6.1. Gap between Deep CNNs Algorithm Application and MPs Imaging Technology

In order to acquire relevant metric data information of microplastics (1 μm–5 mm), image segmentation technologies are introduced to recognize the specified MPs image from background images and other particles. Although automated methods for identification and classifying have been very successfully in medical image field, there is still a certain gap in the application of these methods to microplastic imaging due to the size class distribution which is the most sensitive parameters used by artificial intelligence from microplastics.

At this time, the need for new definitions and new methods for the identification, quantification, and characterization of existing microplastics is evident in the high complexity and diversity of studying this, perhaps the most challenging, analyte. In order to meet the quantification method of microplastics, it is necessary to reliably and sensitively identify, quantify and characterize MPs in the whole size range and different media, and establish or improve the existing artificial assumptions, data models and pretreatment from the perspective of artificial intelligence. However, this work requires further in-depth cooperation between artificial intelligence and chemistry, marine science, and environmental science, and enhances interdisciplinary research and cooperation.

Due to the significant differences in microplastics dimensions (1 μm–5 mm), there is no one-size-fits-all solution for microplastics. In terms of identification and classifying and counting on marine debris pollution in clusters #3–#5 from Figure 25, these methods often meet some issues with the classification of irregular object patterns and a relatively large noise background and interference signal, such as chain-aggregated, convex-shaped, and noise. Additionally, some of these algorithms rely on too much human assumptions and various time-consuming data prepossessing, which is still in a semi-artificial and semi-intelligent state, it is very inconvenient to detect and identify specified shapes of microplastics. In the context of microplastics classification, the pixels of the particles must be obtained to estimate their size. In this case, more deep-learning segmentation architectures (such as FCNN, Mask-RCNN, and U-Net) can be used, because they can not only detect objects, but also label each pixel with the class of objects around it. Techniques for automatically counting and classifying microplastic particles between 1 μm and 5 mm in size have not been widely studied and could stimulate the development of new analytical methods in a number of interdisciplinary fields.

With the application of advanced machine learning, especially deep learning, such as GoogLeNet, ResNet, and U-net, deep convolutional neural networks (CNNs) have once again replaced traditional ML algorithms in a wide variety of research fields. Deep learning networks have approached or surpassed humans in some specific tasks, but there is still no one-size-fits-all algorithm that can solve all problems. In solving a certain class of problems, it seems to be extremely important to choose and customize appropriate model and algorithm to solve a given problem. To some certain more depth CNNs model is designed when it is engaged into a complicated method for a specific problem. Unfortunately, the more depth the deep CNNs model algorithm designs, the more parameters the model needs. Therefore, it is hard for the interpretability of the model and how to find tradeoff between performance and interpretability for deep CNNs model is a critical point. In other words, it is worth noting that the lack of transparency for a deep CNNs algorithm may make the method unacceptable and the balance strategy between the interpretability and performance is a critical thinking for constructing a deep CNNs model.

### 6.2. Data Open Access and Cooperation Mechanisms

In order to be able to train deep neural networks, there is an increasingly strong demand for large-scale real datasets with a large number of manually annotated results. From the publications of WoS in the past five years in the AI-based MPs imaging field, although deep CNNs models have been already introduced into different MPs environmental fields, there are still highly demanding for long-term data sharing and co-operation mechanisms from different institutes. In terms of microbial communities on marine debris pollution in clusters #1 from Figure 25, with the help of advanced machine learning approaches the existing deep learning methods coupled with data set, such as ChemIDplus and ToxCast, published and shared images database can be optimized and refactored based on the available intellective base. However, the most possible obstacle is the lack of related original data which cannot be accessible from different organizations or institutes so that the full potential of deep CNNs models is restricted by limited data open access. Furthermore, most productive authors are always from the same institute and have the closest co-operation relationship internally from Figure 10, Figure 11 and Figure 12. Thus, the severe situation made the raw data sharing become dramatically worse. From the unbalance collaboration network of countries/regions in Figure 8, the developed countries need to improve the widespread data open-access with developing countries, such as South-America and Africa, and help more institutes and countries co-operate in date set, artificial intelligence, information sharing, etc., to fill in the gaps and deficiencies for the above knowledge and solve the “data island” and the current situation of working behind closed doors.

### 6.3. Demand for Automatic Processing Innovation of the High-Volume Imagery

There are some challenges in huge volume of anthropogenic plastic debris. Therefore, there are still major bottlenecks in the estimation of efficient access to beach litter. With the help of advanced UAV technology, smart robot technology, etc., techniques for processing high-throughput image data generated by UAVs were engaged in this research area, which is that the huge volume of images was developed through machine learning and deep learning, applied in identification and classification. To a certain extent, although the machine learning tool encountered too many challenges in correctly identifying objects based on the high-resolution images from UAV, such as low image resolution, false positives, inaccuracy, etc., this type of demonstration is still good practice in efficiently acquiring anthropogenic plastic debris which is promising and promotes efforts to further develop new technologies and implement them on a larger scale and scope.

Another burning question is from tremendous amounts and different data types, which posed lots of challenges to the data pre-process. Many types of tremendous MPs imagery are hard to pre-process before coupling with deep CNNs algorithm, such as data scaling. Accordingly, based on different deep CNNs algorithms, tremendous MPs imagery dataset, including different ranges, units and scales should be rescaled to meet the corresponding data standardization and prerequisite. The complexity from huge volume data posed the great challenge, which is the major obstacle on automatic processing of the high-volume imagery.

## 7. Conclusions and Perspectives

Although a large number of data have been published in terms of automatic quantification and interaction with the ecology of anthropogenic plastic debris, there are still many outstanding problems in terms of AI technical limitations, data sharing, data complexity from huge-volume imagery, depth and interpretability of algorithmic models, and cooperation features, etc. To realize a wider use of advanced machine learning approaches in microplastic-imaging fields, there are still various challenges waiting for solutions to fill these technology and knowledge gaps in future. Among them, exploring efficient and low-cost automatic quantification of microplastics in images by their physical characterizing properties is the most urgent and important. It is worth noting that the problems of microplastics water-soluble synthetic polymers and interaction with other ecology environments are critical to broaden its the depth of investigation with the help of deep learning technologies. Looking for efficient artificial intelligence algorithms and rapidly quantification methods, even including intelligent robot technology, smart UAV, is important to reduce time-consuming and promote its application and efficiency in microplastics imaging. Moreover, the experimental study on the creating and sharing robust data set, such as spectral libraries and toxicity databases, is still at the early stage, which are still needed to strengthen cooperation and sharing between different countries and institutes. It is also important to optimize and refactor the existing deep learning models and interpretability of deep CNNs model based on the available intellective base. Thus, the corresponding experimental and theoretical work should be further carried out in the intersection of artificial intelligence and microplastics in the near future. Again, it is expected that these perspectives will provide some insight in the future to help reduce and minimize the impact of microplastics on the environment and human health.

## Figures and Tables

**Figure 1 ijerph-20-01150-f001:**
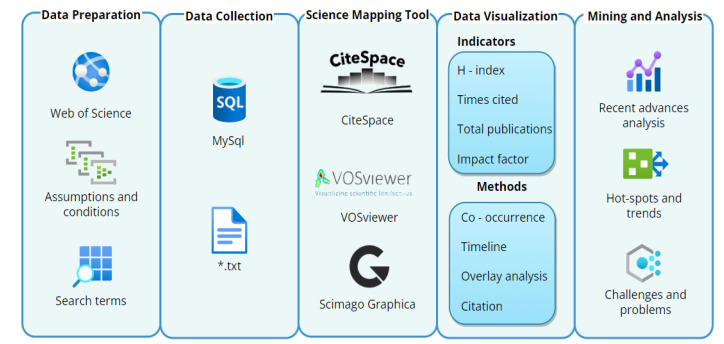
Whole framework of science mapping analysis.

**Figure 2 ijerph-20-01150-f002:**
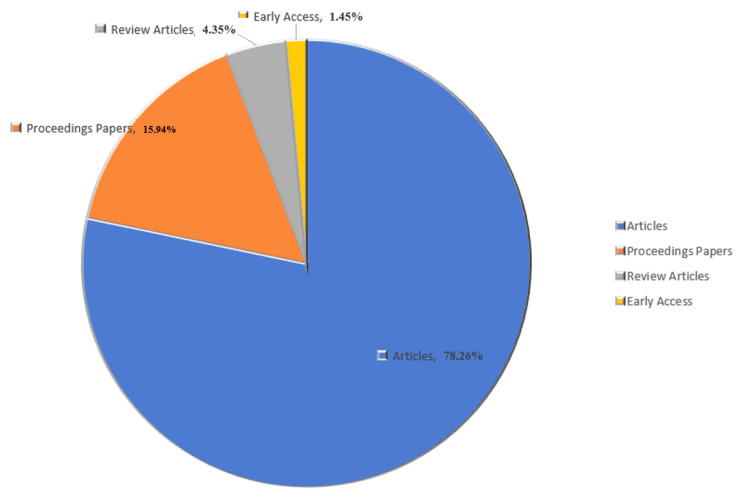
Types of all 69 documents.

**Figure 3 ijerph-20-01150-f003:**
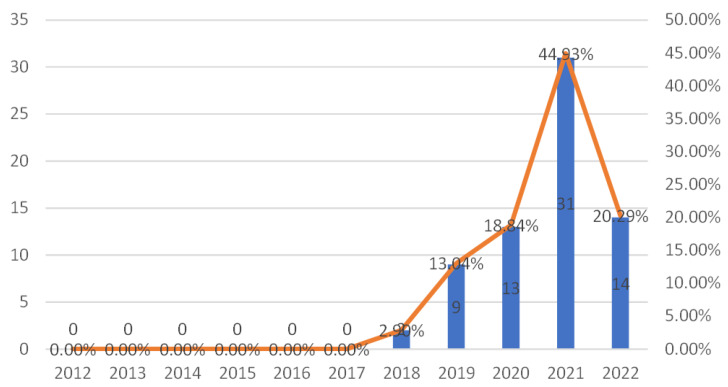
The number and proportion of published papers per year since 2012.

**Figure 4 ijerph-20-01150-f004:**
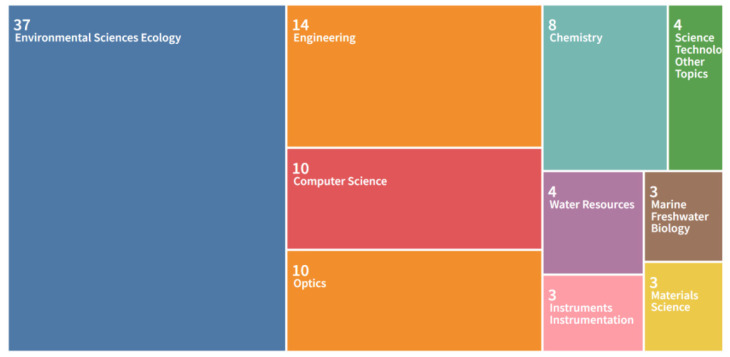
Top 10 hot research areas (generated using Web of Science on data).

**Figure 5 ijerph-20-01150-f005:**
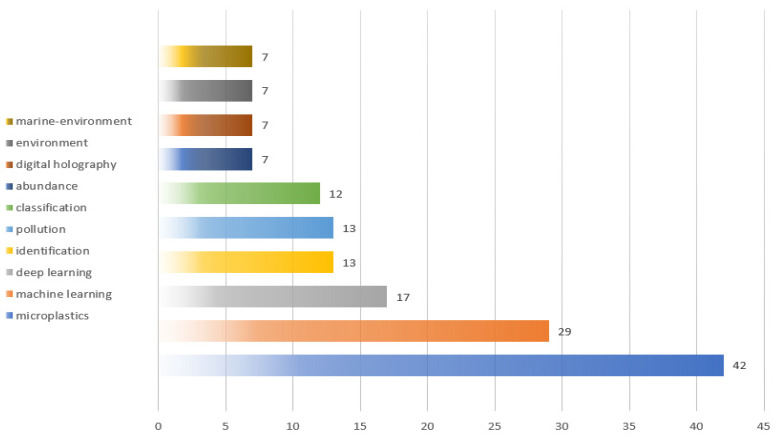
Top 10 high frequency keywords.

**Figure 6 ijerph-20-01150-f006:**
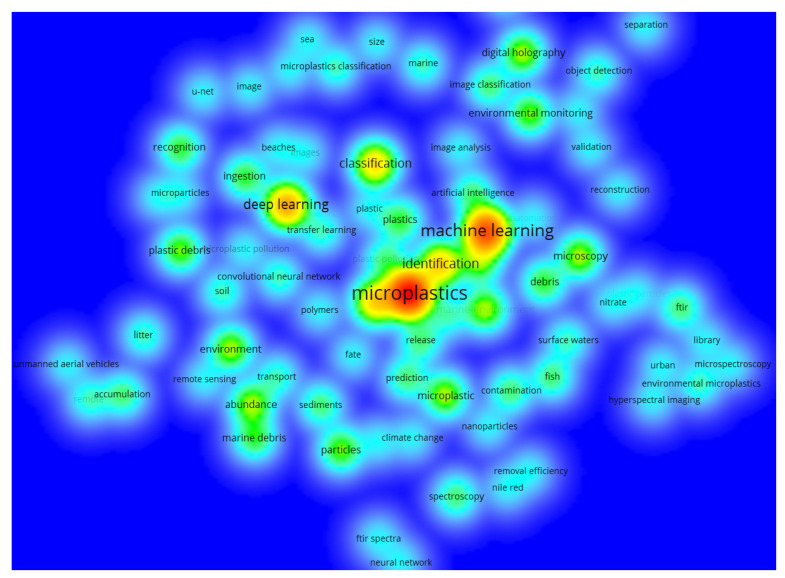
Themes of all 69 documents (visualized with VOSviewer).

**Figure 7 ijerph-20-01150-f007:**
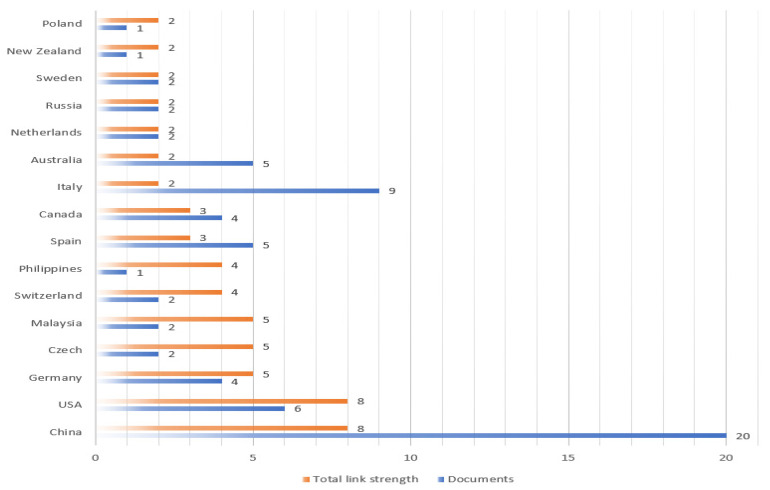
The most active countries/regions.

**Figure 8 ijerph-20-01150-f008:**
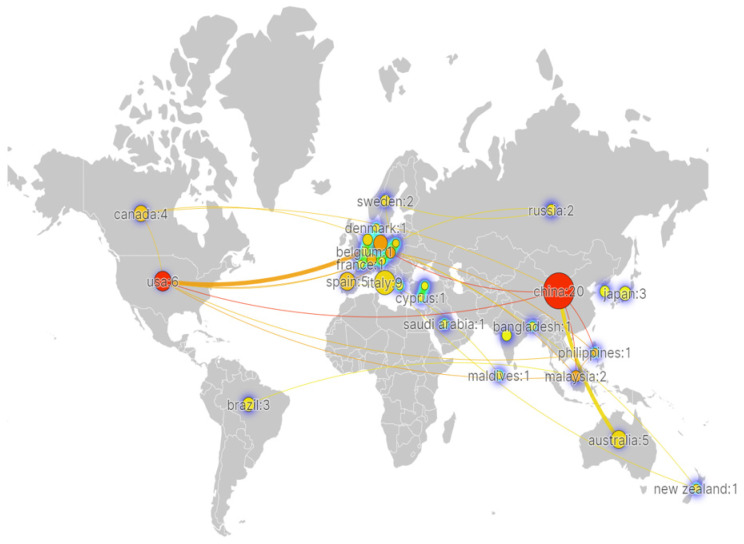
Collaboration network of countries/regions (visualized with Scimago Graphica).

**Figure 9 ijerph-20-01150-f009:**
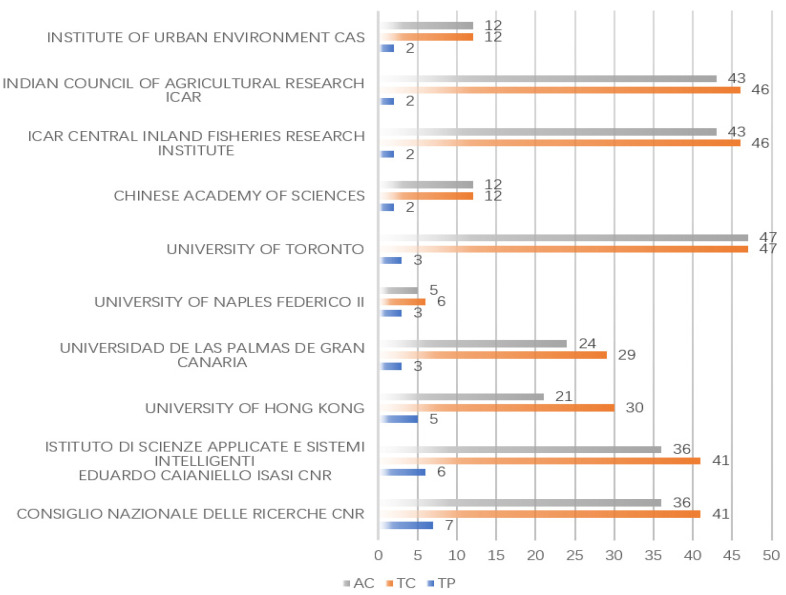
The most contributing affiliations.

**Figure 10 ijerph-20-01150-f010:**
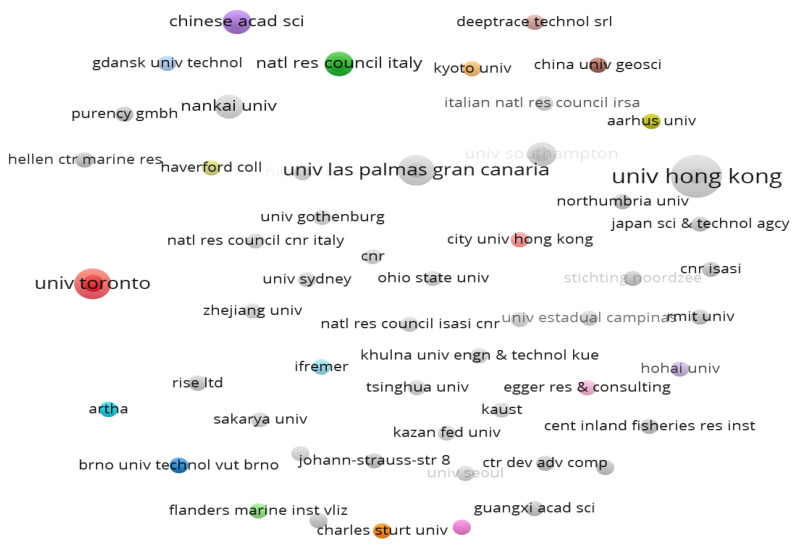
Collaboration networks of all 133 institutes (visualized with VOSviewer).

**Figure 11 ijerph-20-01150-f011:**
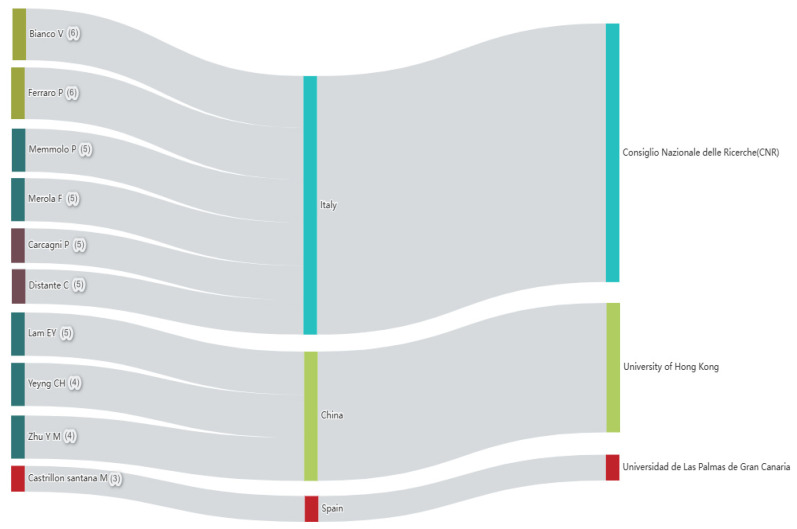
The most active authors from 2018 to 2022.

**Figure 12 ijerph-20-01150-f012:**
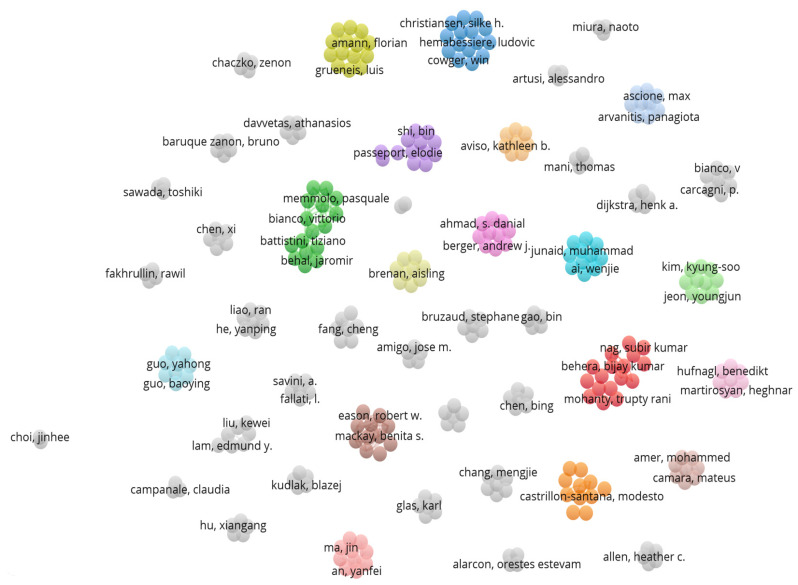
Closest collaborative relationship of authors (visualized with VOSviewer) by setting the threshold to 1.

**Figure 13 ijerph-20-01150-f013:**
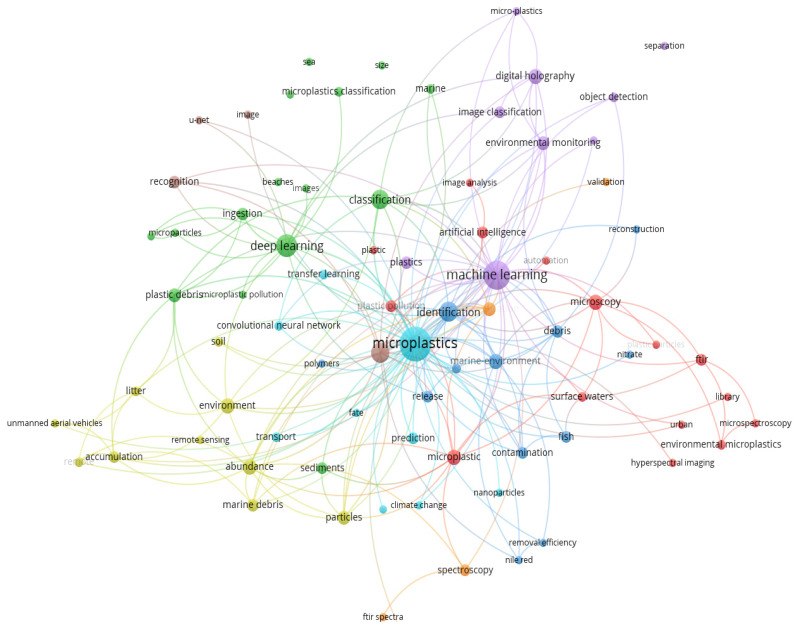
Keyword co-occurrence network structure (visualized with VOSviewer on data).

**Figure 14 ijerph-20-01150-f014:**
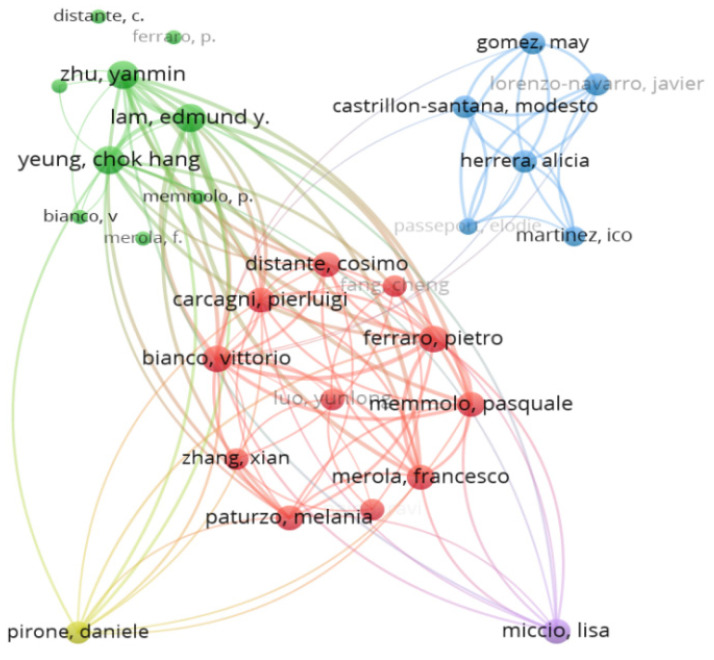
Cited author network structure (Threshold = 2) (visualized with VOSviewer).

**Figure 15 ijerph-20-01150-f015:**
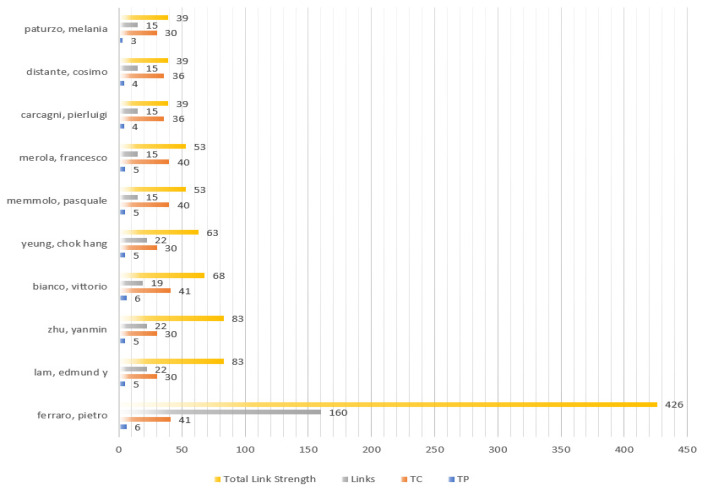
The most active citation authors.

**Figure 16 ijerph-20-01150-f016:**
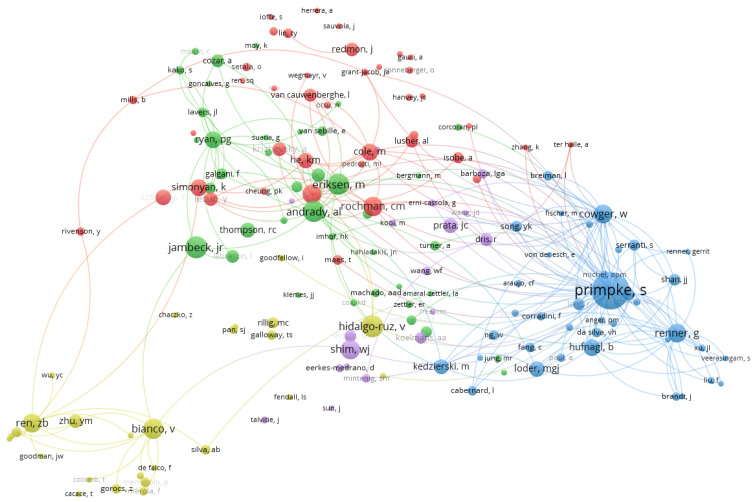
Co-cited network of authors (Threshold = 4) (visualized with VOSviewer).

**Figure 17 ijerph-20-01150-f017:**
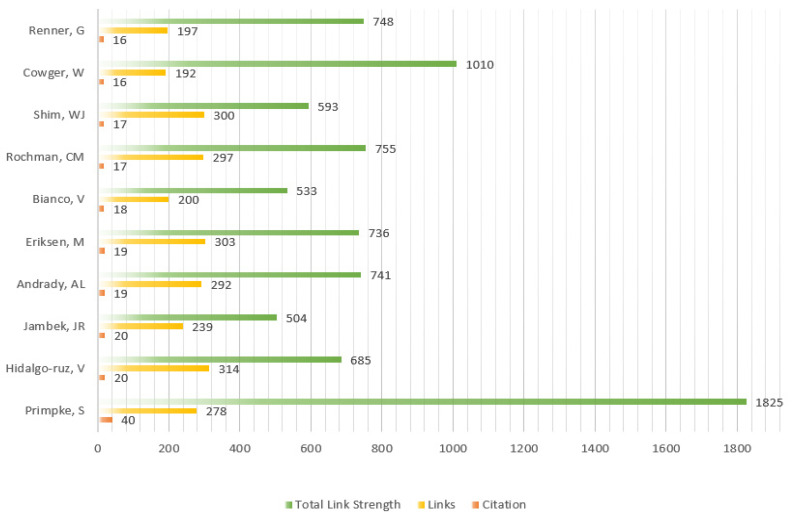
The most co-citation authors.

**Figure 18 ijerph-20-01150-f018:**
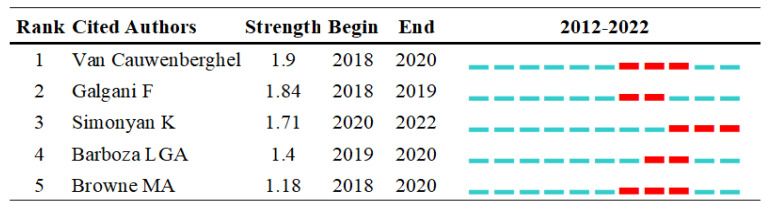
The most co-cited authors with citation bursts (Visualized with CiteSpace. The red bar: the year of the citation burst, and the light blue color: no citation burst.).

**Figure 19 ijerph-20-01150-f019:**
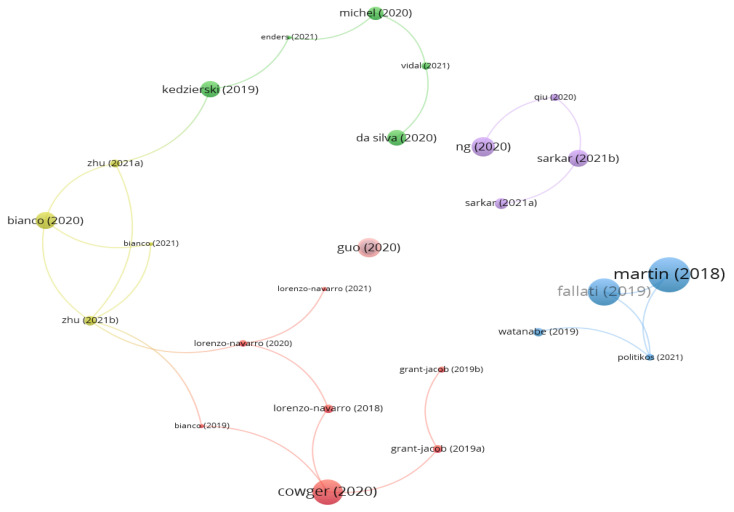
Cited reference network structure (visualized with VOSviewer) (Threshold = 4).

**Figure 20 ijerph-20-01150-f020:**
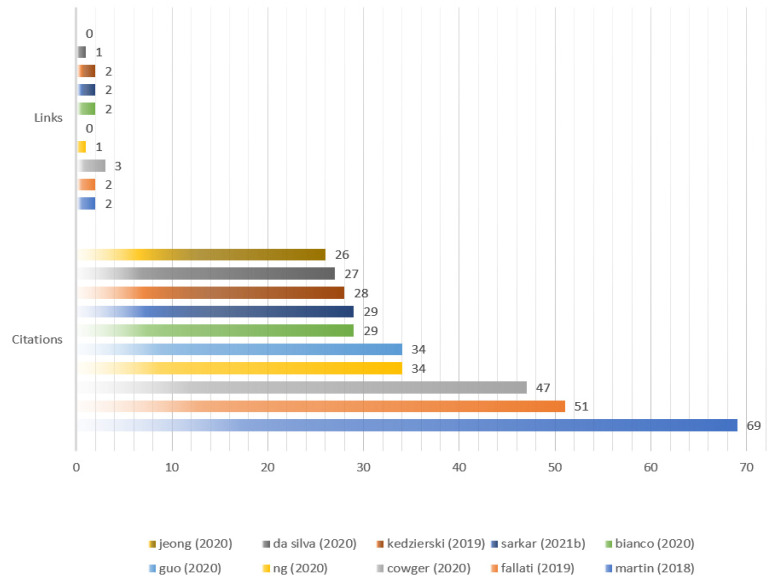
The most cited references.

**Figure 21 ijerph-20-01150-f021:**
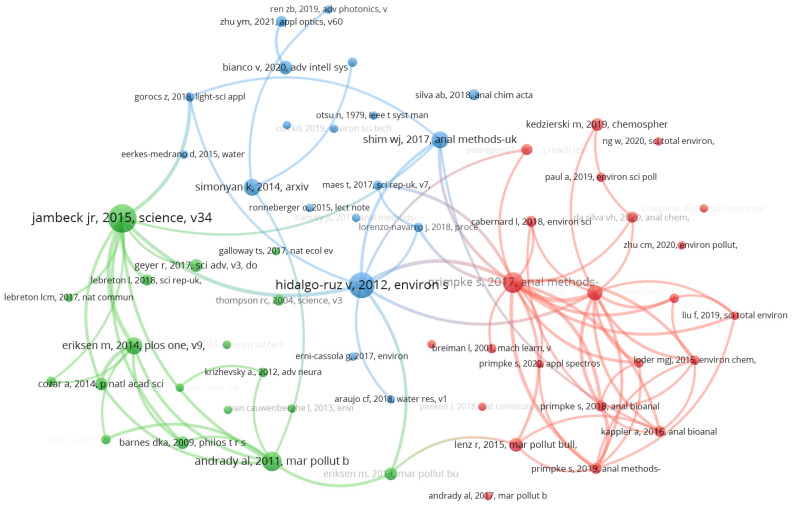
Co-cited reference network (visualized with VOSviewer).

**Figure 22 ijerph-20-01150-f022:**
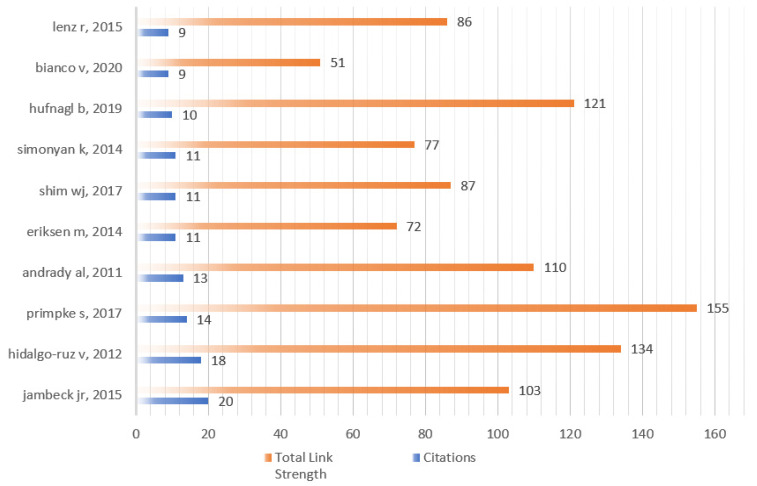
The most co-citation references from 2018 to 2022.

**Figure 23 ijerph-20-01150-f023:**
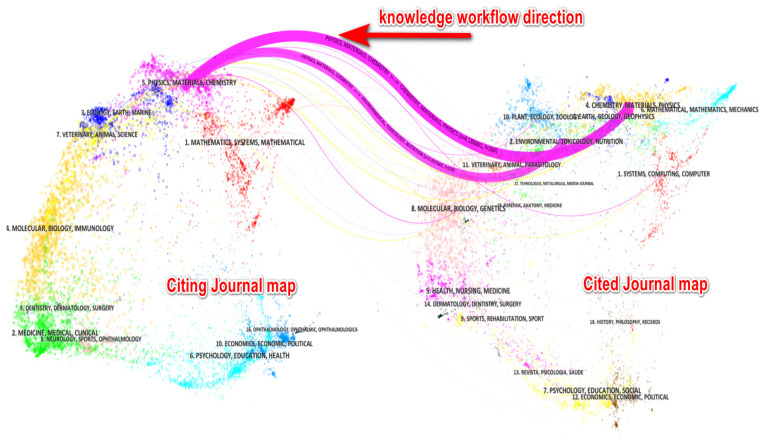
Overlay journals analysis of all 69 publications (generated using CiteSpace on data).

**Figure 24 ijerph-20-01150-f024:**
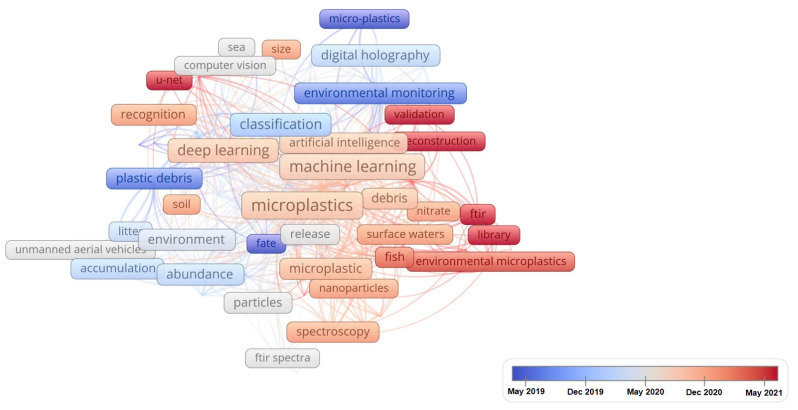
Keywords temporal evolution (visualized with VOSviewer).

**Figure 25 ijerph-20-01150-f025:**
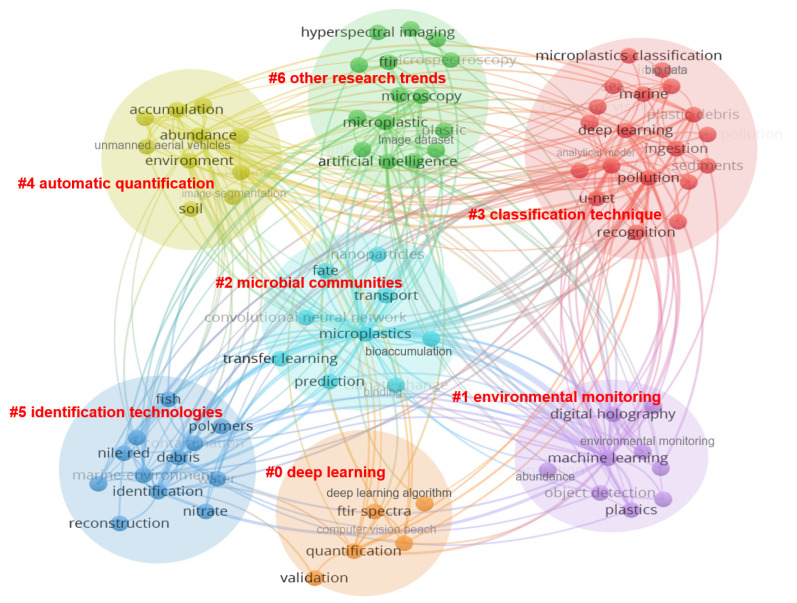
Research hot-spots from cluster view (generated using CiteSpace on data).

**Table 1 ijerph-20-01150-t001:** Search terms.

Prefix	Synonyms	Approaches	Processing
Micro-	Microplasti *	Artificial intelligence	automa *
	Micro-plastic *	convolutional neural network *	calculat *
Nano-	Nanoplastics	deep learning	count *
	Nanometer-plasti *	machine learning	classif *
	MPs	neutral network *	detect *
			digit *
			identif *
			imag *
			quanti *
			recogni *
			sort *
			statist *
			visualiz*

*: a keyword search wildcard.

**Table 2 ijerph-20-01150-t002:** Country names.

Geographical Distribution	Country Name
Mainland China	China
Taiwan	China
Macao	China
Hongkong	China
North Ireland	UK
Scotland	UK
Wales	UK
England	UK

**Table 3 ijerph-20-01150-t003:** Database time-range.

Database	Range
Science Citation Index Expanded (SCI-EXPANDED)	1900—as of now
Social Sciences Citation Index (SSCI)	1900—as of now
Conference Proceedings Citation Index-Science (CPCI-S)	1900—as of now
Conference Proceedings Citation Index-Social Science & Humanities (CPCI-SSH)	1900—as of now
Emerging Sources Citation Index (ESCI)	2017—as of now

## Data Availability

The datasets used or analyzed during the current study are available from the corresponding author on reasonable request.

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
