# Peer review of "A Critical Review on Artificial Intelligence—Based Microplastics Imaging Technology: Recent Advances, Hot-Spots and Challenges"

_ijerph, 2023, doi:10.3390/ijerph20021150_

Round 1

Reviewer 1 Report

the authors have very meticulously examined the scientific publications of the last 5 years on the topic of microplastic detection and impressively present the development and networking of the research field with data sicens methods. it is very interesting to see a meta-analysis of the scientific process in a research field in this form.

the way the paper was structured is comprehensible and follows an internal logic. the selection of the papers is well-founded and methodical. the same applies to the preparation and processing of the information. 

The discussion and conclusions from the prepared graphs are comprehensible and interesting for the reader.

the list of references seems damaged.

Reviewer 2 Report

The authors compiled the literature on artificial intelligence-based microplastics imaging technology. Before publication, there are a few minor suggestions as follows:

 1. Introduction last section needs to be improved

2. The Fig. 4, 6, 7, and 9, 11, 12, 13, need to be improved for the understanding of readers.

3.  It is recommended to be included the future prospect.

Reviewer 3 Report

Dear Yan Zhang and co-authors,

Thank you for your manuscript and your effort in this review. I really appreciate your work but found it very difficult to review your manuscript.

Overall, it is a nice approach with a lot of text and with a high effort. Nevertheless, it is given way too much information, and I also think that some parts are unnecessary, redundant (keywords), or shouldn’t be handled in this way (competitions).

The cluster analysis focusing on hotspots and keywords is, in my opinion, one of the strengths of this study. It gives necessary information about the evolution of keywords, utilization of artificial intelligence, and challenges. In addition, working out of research groups and networks is usable and practicable, I am struggling to find the necessity of publishing it. Moreover, to present the results of the focused countries/regions, and especially between the authors as a competition does not fit to my ethical understanding. I am really concerned about publishing those competitive results with publishing the whole names of authors on top. I found it in general controverse to publish the authors and their records and to put it into a competition together. There is already too much competition in this research field. And with this contribution I really feel uncomfortable, since it increases the competition even more between working groups, research fields, countries, and especially authors. It is clear that all working groups need to cooperate and to collaborate more, especially internationally. Moreover, I also think this is a valuable study showing this big issue, but I would completely exclude names and institutions. A possibility could be to show the cooperation nodes only between countries. Furthermore, to describe it less than a competition, and more like facts and results out of your text mining research.

In the following you will find other points I would like to put your attention to.

-          In general, table descriptions need to be over the table, and figure description must be placed under the figure.

-          Writing names, etc. in capital letters makes it hard to read, and needs to be revised.

-          L41: I did not have the feeling that you focus on your whole manuscript only on particles which are in the size range of 1 mm to 5 mm, and it is the only given size class here. This needs to be clarified in general for your whole manuscript, since the size of objects is one of the most sensitive parameters regarding artificial intelligence, etc.

-          In general, there are a lot of tables and figures in this manuscript – way too many. Moreover, used figures are often redundant, focusing on same topics or are not understandable or readable. For example:

o   Figure 5: Meaning of the size of the nodes is presenting the number of publications, but it is not understandable in the figure itself.

o   Figure 7 and 8 showing almost the same, decide for one or make a combination or put one or both into the SI.

o   Figure 9 seems to be necessary and could replace others!!!

o   L 239: I can totally not understand what is meant here in combination with the table. Would make more sense to show the occurrence in a pie chart or block diagram, if necessary. Like in table 6. Or Figure 4 would be enough, and the other results can only be described in the text.

o   Figure 14: neither good readable, nor informative: thus, it can be easily excluded from the manuscript.

-          Moreover, it seems to me that you tried everything what is possible with the different graphic tools, without focusing on comprehensibility of the figures itself.

-          In addition, using excel, doesn’t fulfill a high standard of a good publication

-          L351: You focusing several times on keywords. I see the point, that there are several things to discuss, but it is just a mirror of your manuscript, that you have way too many information, which is sometimes redundant and/or unnecessary. You already focused on it in line 236 and forward; and at 474 you start again with keywords... Dealing with it in Figure 16 again… Whereas I think that the figure 15 is useful and something new. (dealing with upcoming/evolving keywords in a timeline)

-          Line 494 and following: 5.3. Research hot-spots analysis of author keyword timeline view

o   This is what should be more focused on. The information in Figure 16 needs to be presented in another way since the figure is unreadable and not understandable at all.

-          6. Concluding remarks

It’s way too long, in my opinion, for a concluding paragraph. Thus, I would recommend putting those text part into the discussion part since it is valuable work and rewriting a sharp conclusion. In general, I found the cluster analysis very interesting. You should use this mainly to build your manuscript around it.
